# Prenatal Detection of Congenital Duodenal Obstruction—Impact on Postnatal Care

**DOI:** 10.3390/children9020160

**Published:** 2022-01-26

**Authors:** Kerstin Saalabian, Florian Friedmacher, Till-Martin Theilen, Daniel Keese, Udo Rolle, Stefan Gfroerer

**Affiliations:** 1Department of Pediatric Surgery and Pediatric Urology, University Hospital Frankfurt, 60590 Frankfurt, Germany; kerstin.saalabian@kgu.de (K.S.); Florian.Friedmacher@kgu.de (F.F.); Till-Martin.Theilen@kgu.de (T.-M.T.); Daniel.keese@kgu.de (D.K.); 2Department of Pediatric Surgery, Helios-Klinikum, 13125 Berlin, Germany; stefan.gfoerer@web.de

**Keywords:** congenital duodenal obstruction, prenatal diagnostic, postnatal care, pediatric surgery

## Abstract

Background: Duodenal obstruction is a rare cause of congenital bowel obstruction. Prenatal ultrasound could be suggestive of duodenal atresia if polyhydramnios and the double bubble sign are visible. Prenatal diagnosis should prompt respective prenatal care, including surgery. The aim of this study was to investigate the rate and importance of prenatally diagnosed duodenal obstruction, comparing incomplete and complete duodenal obstruction. Methods: A retrospective, single-center study was performed using data from patients operated on for duodenal obstruction between 2004 and 2019. Prenatal ultrasound findings were obtained from maternal logbooks and directly from the investigating obstetricians. Postnatal data were obtained from electronic charts, including imaging, operative notes and follow-up. Results: A total of 33/64 parents of respective patients agreed to provide information on prenatal diagnostics. In total, 11/15 patients with complete duodenal obstruction and 0/18 patients with incomplete duodenal obstruction showed typical prenatal features. Prenatal diagnosis prompted immediate surgical treatment after birth. Conclusion: Prenatal diagnosis of congenital duodenal obstruction is only achievable in cases of complete congenital duodenal obstruction by sonographic detection of the pathognomonic double bubble sign. Patients with incomplete duodenal obstruction showed no sign of duodenal obstruction on prenatal scans and thus were diagnosed and treated later.

## 1. Introduction

Duodenal atresia is the most common type of congenital duodenal obstruction (CDO) and occurs in 1 per 5000–10,000 live births [1]. Congenital duodenal obstruction could be classified as complete congenital duodenal obstruction (CCDO) and incomplete congenital duodenal obstruction (ICDO). Complete duodenal obstruction usually results from pure bowel atresia or annular pancreas, as opposed to incomplete atresia, which can be caused by an intraluminal web with a central opening. In cases of intestinal malrotation, extrinsic compression through Ladd’s bands can also cause or aggravate duodenal obstruction [1].

The detection of the double bubble sign by prenatal US is highly suspicious for CDO [2]. Recognition of CDO early in pregnancy provides multiple benefits for the patient and the family, including, but not limited to, observed delivery in a tertiary obstetric unit and prompt postnatal treatment, including early surgery. Patients with the prenatal diagnosis of CDO generally show an improved outcome in terms of lower morbidity due to earlier surgery and fewer complications [3].

Duodenal obstruction is suspected to occur around week 12 of pregnancy, possibly due to the failure of recanalization of the duodenal lumen [1]. Annular pancreas is thought to represent aberration in the development of the ventral pancreatic bud [4]. No consensus has been reached on the origin of this anomaly. An influence of the hedgehog signaling pathway is suggested by some studies involving hedgehog knockout mice [5,6]. In some families, several cases of annular pancreas have been reported, thus suggesting a genetic basis for this anomaly [7].

The average time of prenatal diagnosis of CDO has been reported late in pregnancy at around 31 weeks of gestation [3]. Theories suggest that an earlier detection in pregnancy is not possible due to immature gastric emptying pressure preventing dilation of the duodenum [8].

Bishop et al. (2020) reported a mean gestational age for positive ultrasound suggestive of duodenal atresia of 29 + 5/7 weeks (ranging from 19 + 6/7 to 38 + 4/7) [9].

There are sparse data on the criteria for prenatal detection of CDO. Apart from the double bubble sign, polyhydramnios can be a sign of high bowel atresia due to the lack of re-absorption of amniotic fluid [10]. Maternal hydramnios is thought to serve as an aid in the diagnosis of duodenal obstruction [11]; however, the presence of polyhydramnios varies greatly in the literature (from 32% to 81%) [12]. Bittencourt and colleagues reported an overall prenatal detection rate of 44% by findings of polyhydramnios and double bubble sign on antenatal ultrasound exams [3].

Overall, there is very little information on the detection rate of duodenal obstruction in routine ultrasound. Furthermore, even fewer data are available on the prenatal diagnosis of ICDO.

This prompted us to retrospectively examine the results of the prenatal ultrasound scans of women whose children were diagnosed with ICDO or CCDO and had corrective surgery in our center between 2004 and 2019.

## 2. Materials and Methods

After approval of the study by the local institutional review board committee “Ethikkommission des Fachbereichs Medizin der Goethe Universitaet, Frankfurt” (project identification code 261/18; date of approval 26 September 2019), we reviewed all prenatal ultrasound (US) exams of children operated on for congenital duodenal obstruction (CDO) in the pediatric surgery department at our institution between 1 January 2004 and 31 May 2019 in a retrospective study.

The US findings were analyzed for amniotic fluid, abdominal circumference, and visualization of a double bubble sign, as well as abnormalities noted by the examining obstetrician.

The US findings were taken from the maternity logbooks that all expecting mothers in German obstetric care receive from their gynecologist. We included all findings of the three consecutive obstetric ultrasound examinations, which are suggested according to the routine obstetric ultrasound protocol in Germany (first US: 8th + 0–11 + 6th week of gestation, second US: 18 + 0th to 21 + 6th week of gestation, third US: 28th to 31st + 6 week of gestation) [13]. In addition to these routine US scans, mothers are referred to an US specialist (qualified examiner by the German Society for Ultrasound in Medicine (DEGUM: Deutsche Gesellschaft für Ultraschall in der Medizin)) for a prenatal organ screening by recommendation of the attending physician. These US organ screenings were included in our analysis as well.

In alignment with the timing of the routine obstetric sonograms, US findings were grouped by three time periods during pregnancy (first period: 0 to 17 weeks of gestation, second period: 18 to 27 weeks of gestation, third period: 28 weeks to end of pregnancy). None of the outcome parameters (amniotic fluid, abdominal circumference, and visualization of a double bubble sign) were seen before gestational week 17. Therefore, the period up to gestational week 17 was excluded from further analysis.

If there was more than one ultrasound examination conducted in the allotted period, the latest measurement for the examined value was considered in the study.

### 2.1. Amniotic Fluid

The amount of amniotic fluid (AF) was analyzed, and the patients were subdivided by polyhydramnios, oligohydramnios, normal amount of amniotic fluid or control examination needed.

### 2.2. Abdominal Circumference

The abdominal circumference (AC) was regularly measured during the examinations and documented. If the AC was not given, but the abdominal transversal diameter (ATD) and abdominal anterior–posterior diameter (AAD) were measured, AC was calculated using this formula:AC = 3.14 × (ATD + AAD)/2.

The measurements were compared with the corresponding fetal biometry at the given gestational week using the fetal biometry chart [14].

### 2.3. Double Bubble

If the finding of double bubble was documented once, it was regarded as a diagnosis of duodenal atresia, even though in some consecutive exams this finding was not necessarily described any more. Nonetheless, it was evident from the course of action taken by the examiners (birth in specialized center and information of the parents) that the diagnosis of duodenal atresia was made.

### 2.4. Statistics

The median was calculated for (gestational) age and birth weight of the patients. The frequency of US findings for amniotic fluid, abdominal circumference and double bubble sign was compared between patients with ICDO and CCDO. To test for differences, the results were analyzed by the Fisher’s exact test and the Wilcoxon–Mann–Whitney *U* test. *p*-values < 0.05 were regarded as statistically significant. For statistical analysis, IBM SPSS Statistics Version 2019 was used.

## 3. Results

We identified 64 patients operated on for congenital duodenal obstruction in our institution. In total, 33 parents of patients with duodenal atresia (52%) agreed to supply information on prenatal ultrasound scans. A total of 15 patients (45%) had a complete duodenal obstruction and 18 patients (55%) suffered from incomplete duodenal obstruction. The patients in the CCDO group were generally born with a lower birth weight and younger gestational age. None of the incomplete duodenal obstructions were prenatally detected whereas 11/15 (73%) of the complete duodenal obstructions were diagnosed before birth. This results in an overall detection rate of 33.3% (11 out of 33 included patients with CDO). The median age at prenatal diagnosis was 25.3 weeks of pregnancy (18–33.1) and the diagnosis was always obtained through the visualization of double bubble during ultrasound (Table 1).

### 3.1. Amniotic Fluid

The median gestational age at first sonographic detection of polyhydramnios was 31.6 weeks (range 28.1–38.6). No polyhydramnios was detected during the scan between the 18th and 28th week of gestational age. During the second scan, polyhydramnios was only detected in fetuses with CCDO (5/15 patients, 33%) and no polyhydramnios was seen in fetuses with ICDO (*p* = 0.013) (Table 2).

### 3.2. Abdominal Circumference

Abdominal circumference values were grouped in percentiles according to the corresponding fetal biometry chart [14]. Patients with complete and incomplete CDO showed a normal distribution of abdominal circumference according to their gestational week, and there was no significant difference between abdominal circumferences in patients with complete and incomplete duodenal obstruction (Table 3).

### 3.3. Double Bubble

The double bubble sign was only detected in fetuses with CCDO and not in fetuses with ICDO (11/15 (73.3%) vs. 0/18 (0%), *p* < 0.001). The double bubble sign was detected in these fetuses at as early as 18.0 weeks of gestation at a median age of 25.3 weeks of gestation (range 18.0–33.1). No double bubble sign was seen during the first routine ultrasound before the 18th week of gestation.

In four patients (26.7%) with CCDO, a double bubble sign was already detected during the second routine US pregnancy check-up at a median age of 20.2 weeks of gestation (range 18.0–26.7). In seven additional patients (46.7%), the double bubble sign was visible for the first time during the third time point of the routine US pregnancy check-up at a median age of 32.4 weeks (26.7–37.9) (Table 4).

In total, 7 of 33 (21.2%) patients were sent for a special, so-called “organ ultrasound”, by a DEGUM-certified practitioner. In this exam, the detection rate of CCDO was 75% (3/4) and 0% (0/3) for ICDO (Table 5). A total of 2 out of the 3 patients in which duodenal atresia was diagnosed were already suspected to have duodenal obstruction after the standard US scan by their primary OBGYN. The third patient was sent for a DEGUM qualified ultrasound by the attending OBGYN to further investigate a suspicious heart ultrasound and an ASD was confirmed during the examination. CCDO was not reported.

### 3.4. Postnatal Course

The median age for surgery was 1 day (1–7) in the CCDO group and 62 days (5–4387) in the ICDO group, thus showing significantly earlier corrective surgery in the CCDO group in our cohort (*p* < 0.001).

In the CCDO group itself, the patients who were prenatally diagnosed underwent surgery on day 1 (1–2) and the ones who were postnatally diagnosed went into surgery on day 4 (2–7), thus showing that antenatal diagnosis allowed for earlier surgical correction in our study population (*p* = 0.002).

## 4. Discussion

The first case of prenatal detection of congenital duodenal obstruction by ultrasound was reported by Houlton et al. and Loveday et al. in 1974 [15,16]. Since then, it has been shown that prenatal detection of a congenital duodenal obstruction reduces delayed diagnosis and thus reduces morbidity such as dehydration and disruption of acid–base metabolism. Furthermore, early detection of this malformation provides time for the parents and families to prepare for the time after birth [3].

The youngest patient diagnosed with CCDO in our cohort was diagnosed at the age of 18 weeks of gestation. Balcar et al. described the earliest diagnosis of duodenal atresia in week 22.5 and Miro et al. describe a suspicious case that arose in week 20 but was only confirmed in week 35 [17,18]. Thus, our review describes one of the earliest diagnoses of duodenal atresia by double bubble in the literature. Considering that the fetus starts to swallow amniotic fluid at around week 15 of gestation, the detection of a double bubble sign may therefore be possible from that time on.

Choudhry et al. reported that the majority of their duodenal atresia cases were diagnosed during routine ultrasound screening at 20 weeks of gestation. A large study from China revealed that fewer than half of the patients with double bubble were diagnosed before 24 weeks of gestation [19].

In review of this series of 33 pregnancies, the prenatal diagnosis of CDO was achieved only in cases with complete duodenal obstruction. No patient with ICDO was diagnosed prenatally. This resulted in an overall detection rate of 33% in our cohort. In the subset of patients with CCDO, however, prenatal diagnosis was made in 73.3% of cases. In the literature, the prenatal detection rates of duodenal atresia vary between 44% and 81.4% [1,20,21,22,23]. The authors of these publications, however, did not specify whether patients suffered from CCDO or ICDO.

Interestingly, the patients with CCDO had a lower gestational age and lower birth weight compared with patients with ICDO. In another study by our institution, the same results concerning younger gestational age at birth in the CCDO group were seen [24], but to our knowledge there is no additional information in the current literature about differences in gestational age and weight between those two groups, so further investigations would be required to assess those findings.

Overall, the ultrasound finding of a double bubble sign was the only US finding which led to the diagnosis of CDO in our cohort. We could not detect any additional reliable marker to prenatally diagnose incomplete duodenal obstructions, which explains why to date there are no data reported of antenatal diagnosed duodenal stenosis [25]. The parameters previously thought to indirectly indicate duodenal obstruction, such as polyhydramnios and high abdominal circumference, did not show any diagnostic value for ICDO in our study. Abdominal circumference measurements of the fetuses with both ICDO and CCDO were within the standard distribution and thus provide no diagnostic value. This is supported by McCormick et al., who prospectively examined the finding of an unusually large stomach in routine prenatal ultrasound. They found that only 0.62% of isolated large fetal stomachs pointed to gastrointestinal anomalies, thus concluding that an isolated large stomach seems to be an incidental finding not suggestive of intestinal anomaly [26].

The US findings of the specialized organ US exam by DEGUM-qualified practitioners show an even higher detection rate of double bubble in the CCDO cohort but uniformly miss the diagnosis of ICDO (0/3). One can conclude that there may not be any visible signs pointing to ICDO on prenatal ultrasound scans if highly specialized ultrasound examiners are equally unable to obtain a prenatal diagnosis for this pathology.

## 5. Limitations

The retrospective nature of this study, reviewing US findings of multiple different US examiners, puts a limitation on data interpretation. Although CDO is a rare disease, our study of 33 patients with CDO is a report on a small cohort. Most of our patients are situated in and around Frankfurt am Main, which provides easy access to highly specialized ultrasound examiners. Similarly, the German healthcare system provides for a very strict examination schedule during pregnancy. That is why our findings might not be transferred to the general population living in the German countryside or in countries with limited access to ultrasound examinations. Moreover, our oldest patient was diagnosed in 2004. Since then, obstetric ultrasound imaging has evolved and its diagnostic value has improved. Former ultrasound techniques might have contributed to a limited detection rate of CDO.

## 6. Conclusions

Prenatal diagnosis of CDO is only achievable in cases of CCDO by sonographic detection of the double bubble sign. In cases of ICDO, no diagnostic signs appeared to detect the anomaly prenatally. Diagnosing CDO before birth will ameliorate immediate postnatal care and patients’ outcomes. Therefore, further studies with bigger patient numbers and standardized US at multiple time points during pregnancy focusing on the gastric and duodenal area are necessary to improve the general understanding and diagnosing of ICDO and CCDO prenatally.

## Figures and Tables

**Table 1 children-09-00160-t001:** Demographics of patients with incomplete and complete duodenal obstruction.

	ICDO *n* = 18	CCDO *n* = 15	*p*-Value
Gender (male:female), *n* (%)	9:9 (50.0:50.0)	8:7 (53.3:46.7)	1.000
Median gestational age at birth in weeks (range)	38.7 (35.0–40.1)	37.3 (35.3–40.0)	0.031
Median weight at birth in grams (range)	3250 (1780–4120)	2660 (1850–3600)	0.024
Additional congenital anomalies, *n* (%)	12 (67.7)	11(73.3)	0.722
- Congenital heart anomalies, *n* (%)	10 (55.6)	11 (73.3)	0.469
- Trisomy 21, *n* (%)	4 (22.2)	5 (33.3)	0.697
- Chromosomal anomaly (other than trisomy 21), *n* (%)	2 * (11.1)	0(0.0)	0.486
- Hypothyroidism, *n* (%)	1 (6.7)	2 (13.3)	0.589
- Spinal anomalies, *n* (%)	0 (0.0)	1 (6.7)	0.469
- Renal anomalies, *n* (%)	0 (0.0)	1 (6.7)	0.212
- Other, *n* (%)	7 ^b^ (38.9)	1 ^a^ (6.7)	0.041
Median gestational age at time point of CDO diagnosis in weeks (range)	-	25.3 (18–33.1)	-
Number of double bubble findings during prenatal US	0 (0.0)	11 (73.3)	<0.001
Median age at corrective surgery in days (range)	62 (5–4387)	1 (1–7)	<0.001

CDO: congenital duodenal obstruction; *: Cornelia de Lange Syndrome, Rett Syndrome; ^a^: Funnel trachea; ^b^: Pes calcaneus bilateral and celiac disease, sleep apnea, ectrodactyly bilateral and glandular hypospadias, patent omphalomesenteric duct, glutaric aciduria type 1, non-specified retarded development, floppy muscle tone and jejunal stenosis.

**Table 2 children-09-00160-t002:** Amniotic fluid assessment in fetuses with complete or incomplete duodenal obstruction by ultrasound scan during two different periods in pregnancy (18 to 28, and ≥28 weeks of gestation).

Pregnancy Time Period	Amniotic Fluid Assessment by US Scan	ICDO (*n* = 18)	CCDO (*n* = 15)	*p*-Value
18–<28 WOG	Median WOG (range)	20.95 (18.6–27.0)	19.6 (18.0–26.4)	0.243
	- Normal, *n* (%)	15 (83.0)	13 (87.0)	1.000
	- Polyhydramnios, *n* (%)	0 (0.0)	0 (0.0)	-
	- Data not available, n (%)	3 (17.0)	1 (7.0)	0.607
	- Control needed, *n* (%)	0 (0.0)	1 (7.0)	0.455
>28 WOG	Median WOG (range)	29.5 (28.0–33.4)	31.6 (28.1–38.6)	0.104
	- Normal, *n* (%)	14 (78.0)	5 (33.0)	0.015
	- Polyhydramnios, *n* (%)	0 (0.0)	5 (33.0)	0.013
	- Data not available, *n* (%)	4 (22.0)	4 (27.0)	1.000
	- Control needed, *n* (%)	0 (0.0)	1 (7.0)	0.455

WOG: weeks of gestation during ultrasound; US: ultrasound.

**Table 3 children-09-00160-t003:** Abdominal circumference assessment in fetuses with complete or incomplete duodenal obstruction by ultrasound scan during two different periods in pregnancy (18 to 28, and ≥28 weeks of gestation).

Pregnancy Time Period	AC Assessment by US Scan	Incomplete CDO (*n* = 18)	Complete CDO (*n* = 15)	*p*-Value
18–<27 WOG	Median WOG (range)	20.6 (18.6–24.7)	19.6 (18.0–26.4)	0.381
	- ≥95th percentile	1 (5.6)	1 (6.7)	1.000
	- 50–<95th percentile	7 (38.9)	6 (40.0)	1.000
	- 5–<50th percentile	6 (33.3)	5 (33.3)	1.000
	- <5th percentile	0 (0.0)	0 (0.0)	-
	- Data not available	4 (22.2)	3 (20.0)	1.000
>28 WOG	Median WOG (range)	29.0 (28.0–32.7)	28.9 (28.1–31.6)	0.954
	- ≥95th percentile	0 (0.0)	1 (6.7)	0.455
	- 50–<95th percentile	4 (22.2)	5 (33.3)	0.665
	- 5–<50th percentile	6 (33.3)	3 (20.0)	0.458
	- <5th percentile	1 (5.6)	0 (0.0)	1.000
	- Data not available	7 (38.9)	6 (40.0)	1.000

AC: abdominal circumference; WOG: weeks of gestation during ultrasound; US: ultrasound. Data are median with range () or frequency (%).

**Table 4 children-09-00160-t004:** Detection of the double bubble sign in fetuses with complete or incomplete duodenal obstruction by ultrasound scan during two different periods in pregnancy (18 to < 28 and ≥28 weeks of gestation).

Pregnancy Time Period	Double Bubble Sign Seen in US Scan	ICDO (*n* = 18)	CCDO (*n* = 15)	*p*-Value
18–<27 WOG	Median WOG (range)	-	20.2 (18.0–26.7)	-
	- yes (%)	0 (0.0)	4 (26.7)	0.033
	- no (%)	18 (100.0)	11 (73.3)	0.033
>28 WOG	Median WOG (range)	-	32.4 (26.7–37.9)	-
	- yes (%)	0 (0.0)	11 (73.3)	0.033
	- no (%)	18 (100.0)	4 (26.7)	0.033

WOG: week of gestation during ultrasound; US: ultrasound.

**Table 5 children-09-00160-t005:** Detection rate of the double bubble sign during ultrasound examinations conducted by specially certified DEGUM examiners.

Pregnancy Time Period	Double Bubble Sign Seen in US Scan	ICDO (*n* = 3)	CCDO (*n* = 4)	*p*-Value
Week 18–<27 WOG	Median WOG (range)	20.1 (19.3-20.9)	20.0 (17.4-21.9)	-
	- yes (%)	0 (0.0)	2 (66.7)	0.400
	- no (%)	2 (100.0)	1 (33.3)	0.400
>28 WOG	Median WOG (range)	33.4 (33.4)	30.0 (30.0)	
	- yes (%)	0 (0.0)	3 (75.0)	0.143
	- no (%)	3 (100.0)	1 (25.0)	0.143

## Data Availability

The data presented in this study are available on request from the corresponding author.

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
