# Peer review of "Prenatal Detection of Congenital Duodenal Obstruction—Impact on Postnatal Care"

_children, 2022, doi:10.3390/children9020160_

Round 1

Reviewer 1 Report

Dear authors,

I appreciate your work. I have some recommendation.

-line 91: please replace "sometimes" with the reason for referring to the US specialist

-line 104-105: please explain the abbreviation AF (amniotic fluid)

-line 132-134 and table 1: the difference for gestational age and birth weight between the two groups is statistically significant (p=0.031 and 0.024); so I would replace the expression "slightly younger" and "lighter". Also, this aspect is not at all discussed in the discussion section; I think it is an important finding of the study.

-line 116-120 and table 4: although the double bubble sign was actually written in the records in only 7 patients after 28 WOG, it does not mean that the sign was not present or not visualized; so I would replace number 7 in the table with number 11. Otherwise, the 46.7% of detecting the double bubble sign after 28 WOG for CCDO can be misleading.

-line 204-206: I agree with the idea of early surgery and the importance of antenatal diagnosis, but you compared 11 patients with 4 patients (statistically significance is hard to tell); I would reformulate.

- line 187 and table 5: I wonder if the patient with CCDO who missed the qualified US examiner diagnosis was or not diagnosed by the primary OBGYN ?; the difference between the antenatal detection rate seems little. 

-please check the reference No. 13

Thank you.

Author Response

Reviewer 1

-line 91: please replace "sometimes" with the reason for referring to the US specialist

Thank you for the request. We have changed the sentence accordingly.

-line 104-105: please explain the abbreviation AF (amniotic fluid)

Thank you for the request. We have explained the abbreviation in the text.

-line 132-134 and table 1: the difference for gestational age and birth weight between the two groups is statistically significant (p=0.031 and 0.024); so I would replace the expression "slightly younger" and "lighter". Also, this aspect is not at all discussed in the discussion section; I think it is an important finding of the study.

Thank you for this valuable comment. We have changed the wording accordingly and discussed this relevant finding in the discussion.

-line 116-120 and table 4: although the double bubble sign was actually written in the records in only 7 patients after 28 WOG, it does not mean that the sign was not present or not visualized; so I would replace number 7 in the table with number 11. Otherwise, the 46.7% of detecting the double bubble sign after 28 WOG for CCDO can be misleading.

Thank you for the comment. We have changed the numbers as suggested.

-line 204-206: I agree with the idea of early surgery and the importance of antenatal diagnosis, but you compared 11 patients with 4 patients (statistically significance is hard to tell); I would reformulate.

Thank you for the comment. We have reformulated accordingly.

- line 187 and table 5: I wonder if the patient with CCDO who missed the qualified US examiner diagnosis was or not diagnosed by the primary OBGYN ?; the difference between the antenatal detection rate seems little. 

Thank you for the request. We have explained the respective patient in the result section.

-please check the reference No. 13

Thank you for the request. We have added the available information.

Reviewer 2 Report

This case review is presented with sufficient and clear background introduction of congenital duodenal obstruction and proper statistical analysis. Only a couple of minor suggestions:

-It is interesting to see the distribution of gestational age for the cases reviewed in this study. In Table 2 to 5, median and range of GA are only shown for the total. The authors highlighted some GA distributions in the text. However, it would be helpful if the authors could also show the complete GA distributions for each of the sub-groups in both time periods in the tables.

-The authors mentioned “2 of the 3 patients in which duodenal atresia was diagnosed were already suspected to have duodenal obstruction after the standard US scan by their primary OBGYN (Table 5).”. Table 5 was cited at the end of this sentence but this information is not found in Table 5. I suggest to move the citation to the end of the previous sentence.

-In Table 5, all percentages should be rounded to the same number of digit.

- Cases from one hospital were reviewed in this study. The authors should discuss how the study population could represent the total population and how likely that results in this study could be extrapolated to the general population. Given the small sample size in this study, conclusions based on this study should be made cautiously.

Author Response

Reviewer 2

-It is interesting to see the distribution of gestational age for the cases reviewed in this study. In Table 2 to 5, median and range of GA are only shown for the total. The authors highlighted some GA distributions in the text. However, it would be helpful if the authors could also show the complete GA distributions for each of the sub-groups in both time periods in the tables.

Thank you for the request. We have added the required information.

-The authors mentioned “2 of the 3 patients in which duodenal atresia was diagnosed were already suspected to have duodenal obstruction after the standard US scan by their primary OBGYN (Table 5).”. Table 5 was cited at the end of this sentence but this information is not found in Table 5. I suggest to move the citation to the end of the previous sentence.

Thank you for the request. We have changed accordingly.

-In Table 5, all percentages should be rounded to the same number of digit.

Thank you for the request. We have changed accordingly.

- Cases from one hospital were reviewed in this study. The authors should discuss how the study population could represent the total population and how likely that results in this study could be extrapolated to the general population. Given the small sample size in this study, conclusions based on this study should be made cautiously.

Thank you for the request. We have added a respective to the limitation section.

Round 2

Reviewer 1 Report

Dear authors,

Thank you for responding my comments. I think the manuscript is ready to be published. 

  • line 196 - please explain the abbreviation ASD (atrial septal defect)
  • please check the reference No. 13